# Open-label, cluster randomised controlled trial and economic evaluation of a brief letter from a GP on unscheduled medical contacts associated with the start of the school year: the PLEASANT trial

Steven A Julious,[1] Michelle J Horspool,[1] Sarah Davis,[2] Matthew Franklin,[2] W Henry Smithson,[3] Paul Norman,[4] Rebecca M Simpson,[1] Heather Elphick,[5] Oscar Bortolami,[1] Cindy Cooper[6]

For numbered affiliations see end of article.

**Correspondence to**
Professor Steven A Julious;
s.a.julious@sheffield.ac.uk

## ABSTRACT

**Background** Asthma is seasonal with peaks in exacerbation rates in school-age children associated with the return to school following the summer vacation. A drop in prescription collection in August is associated with an increase in the number of unscheduled contacts after the school return.

**Objective** To assess whether a public health intervention delivered in general practice reduced unscheduled medical contacts in children with asthma.

**Design** Cluster randomised trial with trial-based economic evaluation. Randomisation was at general practice level, stratified by size of practice. The intervention group received a letter from their general practitioner (GP) in late July outlining the importance of (re) taking asthma medication before the return to school. The control group was usual care.

**Setting** General practices in England and Wales.

**Participants** 12 179 school-age children in 142 general practices (70 randomised to intervention).

**Main outcome** Proportion of children aged 5–16 years who had an unscheduled contact in September. Secondary endpoints included collection of prescriptions in August and medical contacts over 12 months (September–August). Economic endpoints were quality-adjusted life-years gained and health service costs.

**Results** There was no evidence of effect (OR 1.09; 95% CI 0.96 to 1.25 against treatment) on unscheduled contacts in September. The intervention increased the proportion of children collecting a prescription in August by 4% (OR 1.43; 95% CI 1.24 to 1.64). The intervention also reduced the total number of medical contacts between September–August by 5% (incidence ratio 0.95; 95% CI 0.91 to 0.99). The mean reduction in medical contacts informed the health economics analyses. The intervention was estimated to save £36.07 per patient, with a high probability (96.3%) of being cost-saving.

**Conclusions** The intervention succeeded in increasing children collecting prescriptions. It did not reduce unscheduled care in September (the primary outcome), but

## Strengths and limitations of this study

► The evaluation was a highly efficient study design using routine data to evaluate a general practice public health intervention designed with children with asthma and their parents.

► The intervention was simple to implement, had good user acceptability and was cost saving.

► The intervention increased prescription uptake in the month prior to the return to school with 30% more prescriptions collected.

► There was no immediate effect in September, but in the wider time intervals of September to December and September to August, there was evidence of effect with a reduction in the mean number of medical contacts.

► The coding of the outcomes from the routine data was challenging and the assessment of adherence was not possible.

in the year following the intervention, it reduced the total number of medical contacts.

**Trial registration number** ISRCTN03000938; Results.

## INTRODUCTION

Asthma episodes and deaths are known to be seasonal.[1] A number of reports have shown peaks in asthma episodes in school-age children associated with the return to school following the summer vacation.[2–10] Children returning to school are exposed to a variety of novel respiratory insults including allergens and viruses, at a time of changing climactic conditions. It has previously been shown that viral infection and allergen exposure in allergen-sensitised asthmatics are

associated with increased hospital admissions for acute asthma.[11]

Our previous research[12] confirmed the increase in unscheduled medical contacts with children with asthma being approximately twice as likely as controls to have an unscheduled medical contact with their doctor around the time of the return back to school. In the same study, it was found that in August, immediately preceding the return back to school, there were 25% fewer prescriptions for inhaled corticosteroids, compared with July and September. Patients who received a prescription for inhaled corticosteroids were less likely to have an unscheduled medical contact after the return to school.

Little is known about the factors that are associated with the drop in prescriptions in August. Research on adherence to paediatric asthma treatment in general has identified weak beliefs about the necessity of asthma medication as a key reason for non-adherence.[13] Given that asthma symptoms decline in the summer months, this may lead to weaker beliefs about the necessity to take asthma medication. The general practitioner (GP) letter was designed to address this belief by emphasising the importance of (re)taking asthma mediation prior to returning to school.

The current study is a cluster randomised trial to evaluate whether a letter sent from a GP at the start of the summer vacation reminding parents of children with asthma of the necessity of taking their asthma medication before the return back to school. The study evaluated whether the letter reduced unscheduled contacts after the return back to school and increased prescriptions in August.

## RESEARCH AIMS AND OBJECTIVES

The aim of the study was to assess if a general practice delivered public health intervention (a letter sent from the GP to parents/carers of school-age children with asthma) can reduce the number of unscheduled medical contacts per child after the school return.

## METHODS
### Study design

The study was an open-label cluster randomised trial where GP practices were randomised to the intervention or usual care. The study protocol and Health Technology Assessment (HTA) report have been published.[14 15] The effectiveness of the intervention was assessed on the basis of reduced unscheduled medical contacts after the return to school in September and prescription uptake prior in August. The primary study period was 1–30 September 2013 after the return to school. The extended study period was 1 September–31 December 2013, since asthma-related appointments are more frequent in these months for children with asthma. The full follow-up period was 12 calendar months from 1 September 2013 to 31 August 2014. Prescription uptake and scheduled

medical contacts such as asthma reviews were evaluated during the periods August 2013 and August 2013–July 2014, respectively.

A cluster randomised trial was chosen due to the nature of the condition of asthma. Even if the study design was individually randomised, there would have been a need for the study to be randomised by household as siblings are likely to have asthma. A further consideration was that we wished for the intervention to represent possible routine care for future implementation. A practice level intervention would represent this.

The health economic analyses were based on a 12-month period from 1 August 2013 to 31 July 2014. The period starts a month earlier than the evaluation of medical contacts in order to incorporate the cost associated with delivering the intervention including any increase in prescriptions or medical contacts in response to the intervention that occurred during August 2013.

The primary outcome was the proportion of patients who had an unscheduled medical contact in September 2013.

The secondary outcomes evaluated included the number of unscheduled medical contacts in September 2013 and the number and proportion of any medical contacts (scheduled and unscheduled) in the same time interval as well as in the time intervals September–December 2013 and September 2013–August 2014. The analyses of the same outcomes were repeated for the other time intervals.

### Participants

Participants were school-age children with asthma, aged between 4 years and 16 years, registered with a GP. The primary analysis population was the intention-to-treat population (ITT) among children aged between 5 years and 16 years of age.

The choice of the age group of 5–16 years as the primary analysis population is due to the difficulty associated with making a diagnosis of asthma among children below this age.[16 17] Patients aged 4–5 years were analysed separately to those aged 5–16 years and are not included in this paper. Additional analyses were restricted specifically to children who had received a prescription for steroid inhalers in the previous year.

### Interventions

Sites were randomly allocated to either: intervention group: sending out the letter, or control group: standard care (no letter).

The intervention was a letter sent from a GP to the parents/carers of children with asthma reminding them to maintain their children's medication and collect a prescription if they were running low (see online supplementary appendix 1). It also advised that should their child have stopped their medication, it should be resumed as soon as possible.

The letter template was developed based on standard letters already used in general practice and

designed to address beliefs about the necessity of taking asthma medication before the return back to school. The wording of the letter had input from the study team, which includes a GP, health psychologist and consultant respiratory paediatrician. The letter was also discussed in detail at two patient and public events that included school-age children with asthma and their parents.[18–20]

The intervention letters were sent out the week commencing 29 July 2013 to obviate the distraction of planning for family holidays and yet leave enough time for parents and children to renew prescriptions and gain benefit from the medication. The letter and the timing of the letter was decided following discussion with the patients and public involvement (PPI) group.[19]

### Patient involvement
There were three PPI consultation events with children with asthma and their parents. The first consultation event was funded by a grant by National Institute of Health Research (NIHR) Research Design Service for Yorkshire and the Humber prior to submission of the grant application in January 2011.

At this first consultation event, it was agreed that a letter from their practice would be a useful reminder and not seen in any way as intrusive. A draft of the proposed letter was reviewed, and the children fed back that they believed that the letter from their GP should be addressed to their parents rather than to themselves.

The second PPI consultation event was held after the grant was awarded in September 2012.[19] At this meeting, the intervention letter was finalised. The general feeling among the group was that the intervention did not adequately reflect the seriousness of asthma as a health condition. It was felt therefore that there was a danger that the intervention could be ignored by parents or that the information it contained could be forgotten. The letter was amended to reflect this input.

The consultation event also discussed the timing of the intervention, and it was proposed to send the intervention the first week of August. The event also reviewed the lay summary for the study and provided input to the logo for the study.

Two parents also agreed to join the TSC for the study. At the first TSC meeting, it was agreed to bring the timing of the intervention forward by a week to the end of July, as asthma medication has a better chance of working the earlier it is used consistently.

A third PPI consultation event was held after the study had been completed that will be discussed in the Discussion.[21] There is a website where the PPI events are detailed (http://www.sheffield.ac.uk/scharr/sections/dts/ctru/pleasant/ppi, assessed 8 December 2017). There has also been a separate publication on the first two PPI consultation events.[20]

### Ethical approval and research governance
National Health Service permissions to conduct the study was obtained for all the primary care trusts in England and health boards in Wales.

Details of an amendment to the protocol are given in online supplementary appendix 2. The amendment was to extend the follow-up period by 1 month to the end of September 2014.

The trial was registered with the International Standard Randomised Controlled Trial Register (ISRCTN) reference number ISRCTN 03000938.

### Setting
The setting was primary care with the unit of cluster being general practices. Site eligibility required practices to be using the Vision IT software and be part of Clinical Practice Research Datalink (CPRD). Site recruitment was conducted by CPRD and the NIHR Primary Care Research Network with the Preventing and Lessening Exacerbations of Asthma in School-age children Associated with a New Term (PLEASANT) study team.[22]

### CPRD recruitment
A practice recruitment pack, consisting of a detailed study information sheet and an expression of interest form, was sent to all 433 practices contributing to CPRD in England and Wales at the time of recruitment.[22] Practices were also recruited through the primary care research network. Recruitment took place over a 7-month period from January 2013 to July 2013. For these practices to be in the trial, they needed to join the CPRD.

### Randomisation and blinding
After each practice gave verbal consent to participate in the trial, they were randomised to either the intervention or usual care.[22] Randomisation was stratified by size of general practice (ie, the 'list size') to ensure that there was an equal sample size—in terms of number of school-age children with asthma—in each arm of the trial. The randomisation sequence was generated by a statistician based within the Sheffield Clinical Trials Research Unit (CTRU), using a blocked randomisation and allocation concealment was ensured by restricting access to the two CTRU statisticians. Once practices had agreed to participate, their identifier and list size was forwarded to the trial statistician for randomisation to one of the two groups. The randomisation was then revealed to the study manager and research assistant. The study team were unblinded throughout the study but had no access to data until after a statistical analysis plan was developed and had no influence on data capture.

### Data management
Data were collected through the CPRD, which captures the coding for each consultation by staff in the practice. The medical consultations and diagnostic codes were reviewed to determine if each contact was a scheduled contact, such as a medicines review, or an unscheduled contact, such as an acute or an out of hours visit.

An independent GP adjudication panel was established to help in the coding. The adjudication panel met three times and did not have access to the randomisation group when reviewing the data. The adjudication panel reviewed and coded 4600 unique terms into scheduled and unscheduled medical contacts. These terms accounted for 92% of all medical contacts, but 17% of all terms were used in the study. Terms not coded by the adjudication panel were coded as unscheduled. In addition, 7.9% of all contacts did not have any terms to indicate consultation type or diagnosis, and free text was used in the database system to which the study team had no access. The adjudication panel advised to code these contacts as unscheduled.

## Statistical methods
### Analysis populations
The study was designed to detect a difference of 5% in the proportion of children who have an unscheduled medical contact (30% vs 25%) with 90% power and two-sided significance level of 5%, with an intraclass correlation (ICC) of 0.03 to account for clustering. Based on this, we estimated that we required 70 practices per arm. It was anticipated that the sample size of 140 practices would equate to approximately 14 000 school-age children with asthma. We assumed equal cluster sizes in the sample size calculation. Sensitivity analyses indicated that the study was robust to the assumptions made for the ICC as well as to practices not sending the intervention and reducing the observed effect size.[15]

Each of the outcomes were evaluated on for each of each subpopulations: children aged 5–16 years (the primary analysis population) and children aged 5–16 years who have prescriptions for steroid preventers. The analyses were restricted specifically to children who had received a prescription for preventer inhalers in the previous year as this was intended to identify the treatment effect in the population likely to receive most benefit.

The primary analyses of effectiveness were performed on the ITT population. Analyses were also conducted on the per protocol (PP) population. The health economic analyses were based on the PP population. ITT analyses comprised all practices for whom data were obtained for the study period (see number of participants and analysis subsets). There were two criteria for exclusion from PP analyses. First, for practices that did not send intervention as requested the entire practice data were excluded from PP analyses. Second, individual children who were not sent the intervention letter. GPs were given discretion to withhold the letter from any children they believed were unsuitable. In such cases, the individual was excluded from PP analyses.

### Analytical methods
The proportion of children having an unscheduled medical contact was analysed separately for each time period using logistic regression with the individual's age, sex, number of contacts the previous September as covariates, the trial arm (intervention or control) as a fixed effect and the design/cluster effect of general practice as a random effect. The proportion of children having a prescription within each time period was analysed in the same manner. The number of unscheduled medical contacts made in each period by the children as well as the number of prescriptions ordered within a time period was both analysed using a random effects negative binomial model in which the same covariates as above were included.[15]

## Health economic methods
An economic evaluation was undertaken to compare the incremental cost per quality-adjusted life-year (QALY) of the reminder letter versus standard care. The perspective of the analysis was that of the NHS (primary and secondary care resource use based on available CPRD data and associated costs). We assumed the intervention would have no impact on mortality or quality of life (utility) beyond 4 months, and as a result, QALYs were calculated for the 4-month postintervention time period. Costs were calculated for 1-year postintervention to allow for changes in the timing of routine asthma care in response to the intervention to be distinguished from changes in the number of scheduled contacts.

Bootstrapped costs were evaluated 12-month postintervention with 1-year linear regression-based baseline adjustment (BA). Costs for the letter intervention were based on a survey of participating practices that included questions on staff members involved as well as staff time.[23] Full details of the methods used in the economic analysis have been published in a separate paper.[24]

## Trial oversight
A trial steering committee (TSC) was established to give oversight to the study. The TSC consisted of an independent chair (GP), two independent members (academic GP and statistician) and two lay members (parents of children with asthma) along with the principal investigator and key staff within the CTRU (as non-voting members). The role of the TSC was to provide supervision of the protocol and statistical analysis plan and to provide advice on, and monitor progress of, the trial.

## TRIAL RESULTS
### Recruitment and participant flow
The target sample size was 140 GP practices. In total, 142 practices agreed to take part in the study. Recruitment of GP practices was undertaken over a 7-month period, details of which have been published.[22] Of these practices, one (a control group practice with 99 children with asthma) withdrew consent after the start of the study for the data to be extracted and stored by the CPRD (independent of the study); this practice was excluded from all analyses. In total, 70 practices (comprising 5917 individuals) were randomised to the intervention (letter) and 71 practices (6262 individuals) to control. The Consolidated

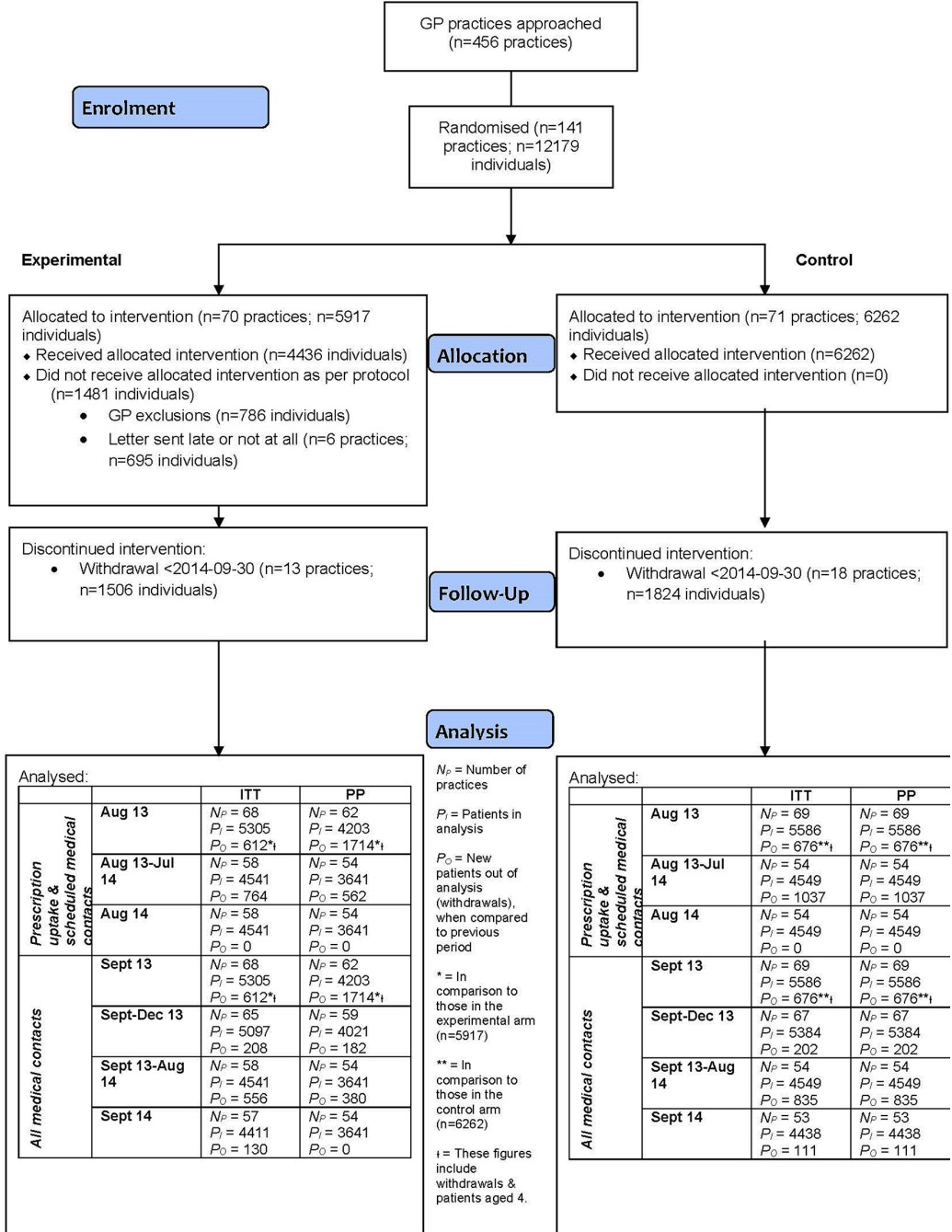

**Figure 1** CONSORT diagram.

Standards of Reporting Trials (CONSORT) diagram is given in figure 1 for the 12-month follow-up of the study.

## Baseline characteristics

The descriptive statistics of the 12 179 subjects and 141 practices are included in table 1A and B. Summaries reported are stratified by intervention type and overall.

## Number of participants and analysis subsets

For each study period, analyses were based only on practices that contributed data to the entirety of that period. In other words, if practices stopped submitting data to CPRD before the end of a given follow-up period, they were excluded from all analyses for that time period. Practices

that changed their software from the Vision IT system were no longer able to participate in CPRD and so withdrew from the study. The data from the practices until the time they withdrew was included in the statistical analysis. Details of the practices within the study during each time period are given in online supplementary appendix 3.

Figure 1 shows the flow of patients and practices for the primary analysis population (aged 5–16 years). Of the 456 practices invited, 433 were through the CPRD and 23 were through the primary care research network and joined the CPRD.[22]

There were 786 GP exclusions in the intervention arm. There were zero GP exclusions in the control arm, as it

**Table 1** Descriptive statistics of patients and surgeries

**(A) Descriptive statistics of sex (frequencies and percentages reported) and age (mean, SD, median, IQR and range reported). Statistics produced at subject level.**

| Variable | Letter (n=5917) | No letter (n=6262) | Total (n=12 179) |
|---|---|---|---|
| Sex | | | |
| Male, n (%) | 3505 (59.24) | 3749 (59.87) | 7254 (59.56) |
| Female, n (%) | 2412 (40.76) | 2513 (40.13) | 4925 (40.44) |
| Age | | | |
| Mean (SD) | 10.51 (3.29) | 10.55 (3.30) | 10.53 (3.30) |
| Median (IQR) | 10.80 (7.88–15.97) | 10.89 (7.80–15.97) | 10.89 (7.80–15.97) |
| Range | 4.05–15.97 | 4.05–15.97 | 4.05–15.97 |
| Unscheduled contacts: September 2012 | | | |
| Mean (SD) | 0.84 (1.20) | 0.88 (1.26) | 0.86 (1.23) |
| Median (IQR) | 0.00 (0.00–1.00) | 0.00 (0.00–1.00) | 0.00 (0.00–1.00) |
| Range | 0.00–10.00 | 0.00–12.00 | 12.00 |
| Unscheduled contacts: September–December 2012 | | | |
| Mean (SD) | 3.65 (3.34) | 3.78 (3.66) | 3.71 (3.51) |
| Median (IQR) | 3.00 (1.00–5.00) | 3.00 (1.00–5.00) | 3.00 (1.00–5.00) |
| Range | 0.00–31.00 | 0.00–51.00 | 0.0 51.00 |

**(B) Descriptive statistics of size (mean, SD, median, IQR and range reported). Statistics produced at surgery level.**

| Variable | Letter (n=70) | No letter (n=71) | Total (n=141) |
|---|---|---|---|
| Size | | | |
| Mean (SD) | 85 (44) | 88 (64) | 86 (55) |
| Median | 80 (49–114) | 75 (41–107) | 76 (45–113) |
| Range | 4–209 | 10–293 | 4–293 |

was impossible for the GPs to exclude individuals from receiving letters when no patients in the control arm were due to receive a letter.

### Clinical results

The primary time point for the analysis was September. Thus, in the primary analysis the proportion of individuals who had at least one unscheduled medical contact in September was 45.2% in the intervention arm, compared with 43.7% in the control arm (adjusted OR=1.09, 95% CI 0.96 to 1.25) (see table 2A). The ICC for the primary analysis was 0.026 and was consistent with the estimate for the sample size calculation. In terms of mean contacts the number of unscheduled contacts are comparable (incidence rate ratio (IRR)=1.02 95% CI 0.94 to 1.12). The results are comparable for children receiving preventer medication (see table 2B).

In the year following the intervention, there was evidence of a reduction in the mean number of medical contacts. As a consequence, the incidence ratio declines as longer time periods are analysed (see table 2A), suggesting that the short-term increase in medical contacts in September is gradually outweighed by decreases in unscheduled contacts in the longer term. There is a non-statistically significant 3% reduction in unscheduled contacts (IRR=0.97 with 95% CI 0.95 to 1.04), and a statistically

significant 5% reduction in the total number of medical contacts (IRR=0.95 with 95% CI 0.91 to 0.99), over the 12 months following the intervention.

Unscheduled medical contacts for children in the trial in the year before and after the intervention are presented in figure 2A. A pronounced drop in unscheduled medical contacts can be seen in August 2012. After the return to school in September 2012 there is an increase in unscheduled medical contacts which peaks in October/November before reducing.

In 2013, there is a similar pattern to 2012 in the control arm. In contrast, in the intervention arm, although there is no immediate effect of the intervention in September, however, the peak in October/November is less pronounced than in the control arm.

The planned analysis was of prescriptions in August. This demonstrated that the intervention (letter) was associated with a statistically significant increase in the uptake of prescriptions in the month of August 2013 (see table 3). In August, 876 (16.5%) patients in the intervention arm had at least one prescription compared with 703 (12.6%) in the control group (adjusted OR 1.43, 95% CI 1.24 to 1.64); the total number of prescriptions was also higher (adjusted IRR 1.31, 95% CI 1.17 to 1.48). In line with the increase in prescriptions, there was also a non-statistically

**Table 2** Analysis of unscheduled and total medical contacts

| | Time period | Treatment arm* Intervention (%) | Control (%) | OR† | 95% CI | Treatment arm* Intervention (mean) | Control (mean) | Incidence ratio† | 95% CI |
|---|---|---|---|---|---|---|---|---|---|
| (A) For all children in the intention-to-treat population | | | | | | | | | |
| Unscheduled | September | 45.2 | 43.7 | 1.09 | 0.96 to 1.25 | 0.81 | 0.81 | 1.02 | 0.94 to 1.12 |
| Contacts | September–December | 80.1 | 79.1 | 1.10 | 0.96 to 1.26 | 3.19 | 3.32 | 0.98 | 0.93 to 1.04 |
| | September–August | 93.1 | 93.3 | 0.97 | 0.82 to 1.15 | 9.08 | 9.37 | 0.97 | 0.95 to 1.04 |
| Total | September | 57.8 | 58.4 | 0.99 | 0.80 to 1.22 | 1.05 | 1.10 | 0.97 | 0.87 to 1.07 |
| Contacts | September–December | 89.3 | 88.4 | 1.06 | 0.89 to 1.27 | 4.31 | 4.43 | 0.95 | 0.90 to 1.02 |
| | September–August | 96.6 | 96.4 | 0.89 | 0.71 to 1.12 | 11.52 | 12.08 | 0.95 | 0.91 to 0.99 |
| (B) For children receiving preventer medication in the intent to treat population | | | | | | | | | |
| Unscheduled | September | 46.3 | 45.4 | 1.07 | 0.94 to 1.23 | 0.83 | 0.84 | 1.01 | 0.92 to 1.10 |
| Contacts | September–December | 81.3 | 81.4 | 1.04 | 0.90 to 1.21 | 3.27 | 3.44 | 0.97 | 0.92 to 1.03 |
| | September–August | 93.9 | 94.6 | 0.84 | 0.69 to 1.02 | 9.31 | 9.71 | 0.98 | 0.92 to 1.14 |
| Total | September | 59.1 | 60.4 | 0.97 | 0.79 to 1.21 | 1.08 | 1.14 | 0.96 | 0.86 to 1.07 |
| Contacts | September–December | 90.4 | 90.5 | 0.98 | 0.81 to 1.19 | 4.43 | 4.70 | 0.95 | 0.89 to 1.01 |
| | September–August | 97.1 | 97.3 | 0.81 | 0.64 to 1.01 | 11.82 | 12.53 | 0.96 | 0.90 to 1.12 |

*The proportions and means are simple summary statistics.
†The ORs and incidence ratios with the corresponding CIs are from a formal statistical analysis allowing for covariates and the effect of clustering.

significant increase in scheduled contacts made in August 2013, in terms of having at least one contact (adjusted OR 1.13, 95% CI 0.84 to 1.52) and a statistically significant increase in terms of the mean number of scheduled contacts (IRR=1.17, 95% CI 1.06 to 1.29).

Preventer prescription collections for children in the trial in the year before and after the intervention are presented in figure 2B. Mirroring the unscheduled contacts in August 2012, there is a reduction in prescriptions collected in this month. After the return to school in September, there is an increase in prescriptions collected with a peak in the interval between October and December followed by a reduction.

In 2013, there is a similar pattern to 2012 in the control arm. In contrast, in the intervention arm, there is a marked increase in prescriptions in August 2013 that appears to continue into September before declining.

Per-protocol population analyses were also conducted, with the results being broadly consistent with the intention-to-treat analyses reported above. However, there were larger effects for the increase in scheduled contacts and uptake of prescriptions in August but smaller effects for unscheduled contacts and total medical contacts.[14]

The analysis of respiratory relation contacts is given in online supplementary appendix 4 and figure 3.

### Health economic results

The full results of the economic evaluation have been published in a separate paper so only key baseline adjusted (BA) base case results are provided here.[24] The average cost per child of sending the intervention was £1.34 per child. The fall in medical contacts over 1 year described in the clinical results led through into the health economic assessment. A mean reduction in costs per child of £36.07 was estimated, and there was 96.3% certainty of the intervention being cost saving. The economic evaluation estimated a mean QALY loss of 0.00017, which is practically zero.

### DISCUSSION

Previous work has shown an increase in the number of unscheduled medical contacts by children in autumn months (September–December), which may be due to the start of the new school term.[12] By sending a letter at the start of the school holidays to remind children of the importance of taking their medication, it was hypothesised that this increase in unscheduled medical contacts may be averted. More specifically, it was predicted that a reminder letter would lead to a greater uptake of inhaler prescriptions in August that, in turn, would lead to increased adherence and, finally, fewer unscheduled medical appointments after the return to school.

There was evidence of an impact on the first part of this pathway as the intervention group demonstrated a higher uptake of prescriptions in August 2013. They also had a non-statistically significant increase in the number of scheduled contacts in the same month.

Data are not available to confirm actual medicine usage and, as a result, it is unclear whether the increased uptake also translated into an increased usage. The original plan was to assess this through the medication possession ratio, which estimates the time a child has collected medication for over the time the child should have collected time

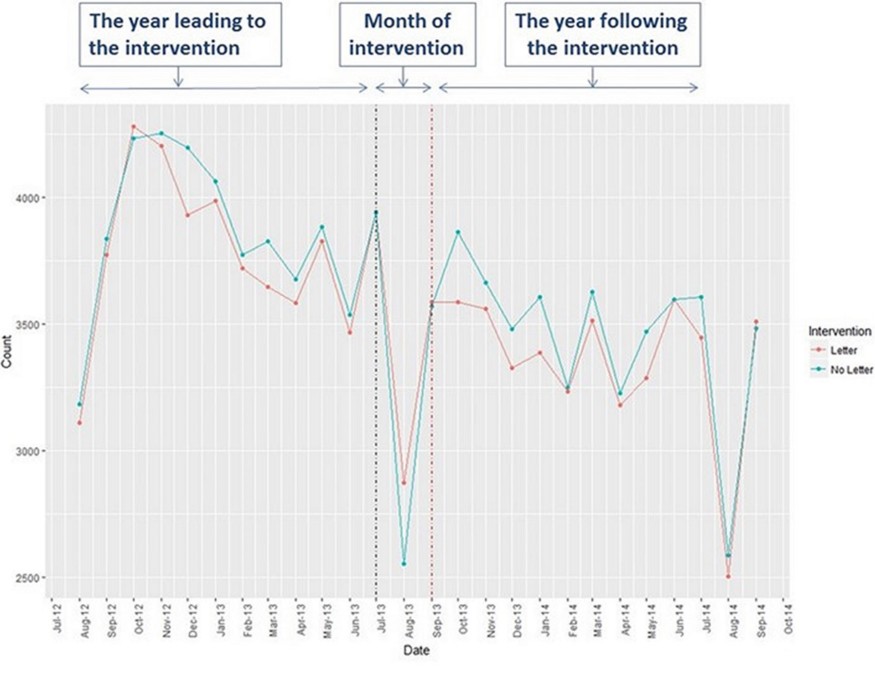

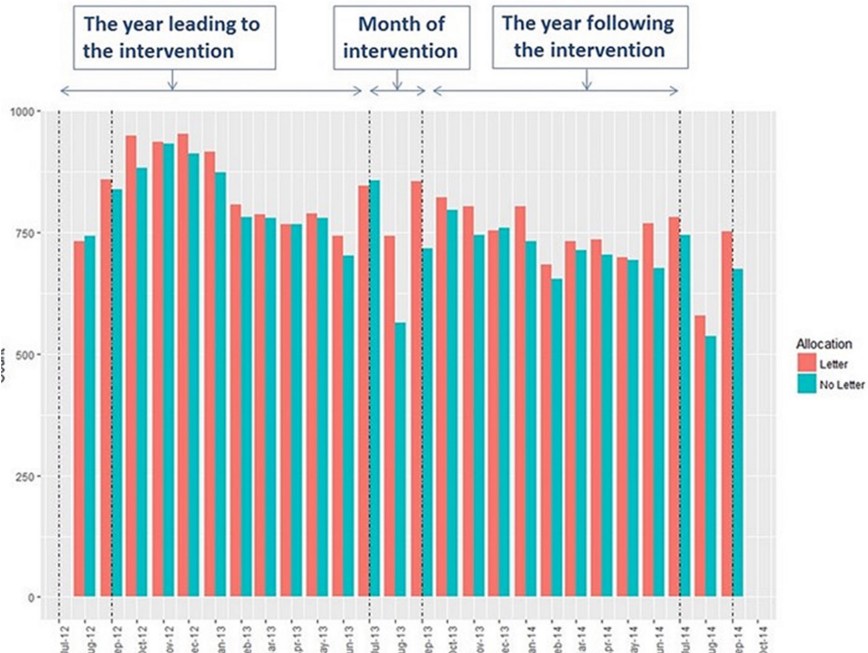

**Figure 2** Unscheduled medical contacts and prescriptions over time.

for. However, this could not be estimated for these data due to inadequate recording of prescription data in the routine data. Further work is required to determine how to assess adherence using such routine data.

The primary endpoint was unscheduled medical contacts in September 2013, which coincided with the start of the new school term. There was no evidence of a reduction in the intervention group. In fact, there was a non-statistically significant increase in the proportion

of children who had an unscheduled medical contact in September.

The increase could be caused by GPs needing to see certain patients before giving a new prescription if they had not had a prescription recently. Evidence to support this is a post hoc observation that for children who had collected a prescription within the last 3 months prior to the start of the study, there was an increase in unscheduled contacts in September with 55.2% of patients in the

**Table 3** Analysis of prescription and scheduled contacts for August

| | Treatment arm* | | | | Treatment arm* | | | |
|---|---|---|---|---|---|---|---|---|
| | Intervention (%) | Control (%) | OR† | 95% CI | Intervention (mean) | Control (mean) | Incidence ratio† | 95% CI |
| **Prescriptions** | | | | | | | | |
| All children | 16.5 | 12.6 | 1.43 | 1.24 to 1.64 | 0.17 | 0.13 | 1.31 | 1.17 to 1.48 |
| Preventer | 17.3 | 13.4 | 1.41 | 1 23 to 1.63 | 0.18 | 0.14 | 1.30 | 1.16 to 1.47 |
| **Scheduled contacts** | | | | | | | | |
| All children | 14.3 | 13.9 | 1.13 | 0.84 to 1.52 | 0.17 | 0.16 | 1.17 | 1.06 to 1.29 |
| Preventer | 14.8 | 14.4 | 1.14 | 0.84 to 1.54 | 0.18 | 0.17 | 1.17 | 1.06 to 1.29 |

*The proportions and means are simple summary statistics.
†The ORs and incidence ratios with the corresponding CIs are from a formal statistical analysis allowing for covariates and the effect of clustering.

intervention arm seeing their GP compared with 54.3% for controls. For patients whose last prescription was 3–6 months prior to the start of the study, the difference between the arms was greater with 42.1% in the intervention arm seeing their GP in September compared with 39.7% for controls.

The way the unscheduled contacts were coded could have also impacted on the outcome. The intervention increased prescription update and collection of a prescription for asthma was coded as an unscheduled medical contact.

A further explanation for the lack of effect of the intervention on unscheduled contacts in September is that September was too early to make an assessment of

efficacy. Given that exposure to infections may take some time to have an impact on asthma symptoms in school-age children, it is possible that making the primary outcome period the first 4 weeks after returning to school was too soon to observe an effect of the intervention.

It is interesting that an effect in favour of the intervention was demonstrated when the measurement period was extended both to December and to the following August. In the extended period, both the total number of contacts per child (ie, scheduled plus unscheduled) and unscheduled contacts were lower in the intervention group than in the control over the extended study period (September–December 2013) and the full year (September 2013–August 2014), although only the effect

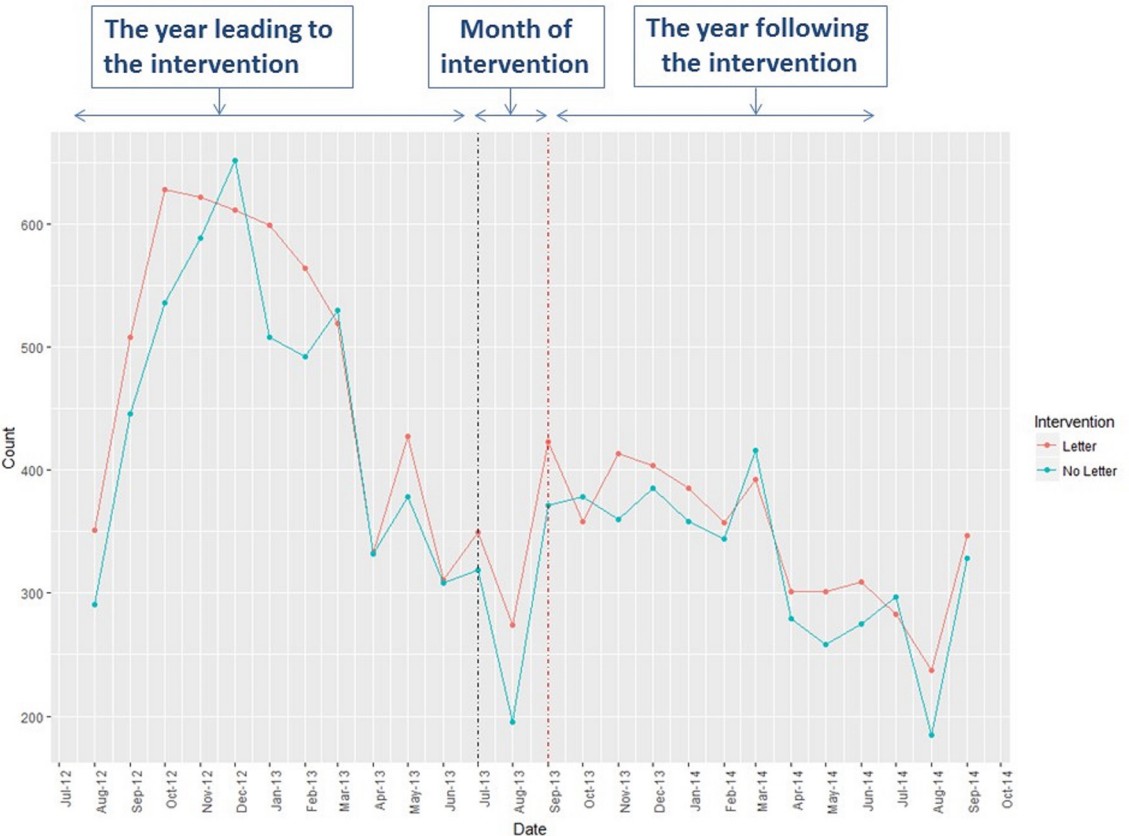

**Figure 3** Unscheduled respiratory medical contacts.

on the total number of medical contacts for the full year was statistically significant.

The effects on medical contacts, though small, are potentially clinically important given the effect the intervention had on prescriptions. The expectation was an increase in prescription update would lead to a reduction in medical contacts. The results are in line with this expectation. Moreover, the minimal cost associated with the intervention meant the intervention was found to have a high probability of being cost-saving overall. With such a relatively low cost intervention, £1.34 per child, and an average cost for an unscheduled surgery visit circa £50, an intervention would only need to reduce the number of contacts by three per year for an average practice with 85 asthmatic children to be cost neutral. The evidence from the trial is that contacts are reduced by approximately 0.6 contacts per child in the 12 months after intervention or 51 per year for an average practice of 85 children.

The economic analysis (which used data over a 12-month period from August 2013 to July 2014) estimated a mean cost saving across the base case of £36.07 per child. So, although the study did not have a significant effect for the primary endpoint, the minimal cost associated with the intervention meant the intervention was found to have a high probability (96.3%) of being cost-saving overall, primarily due it its effect on reducing the total number of medical contacts over the following year. In the UK alone, there are over one million children with asthma. The intervention thus has the potential to provide health service savings if implemented.

The results were discussed with children with asthma and their parents at a PPI consultation event.[21] At the event, attendees felt that the savings per child was an important result and suggested that the impact of the intervention could have been greater if it had been repeated over a number of years. The letter could then assist parents and children as they plan for the school return each year.

There is evidence of good user acceptability with over half the practices who responded to a survey reporting that they had repeated the intervention the year after the study.[23] Once the intervention is set up for 1 year, the costs then associated with sending it out in subsequent years are less given that many of the school-age children with asthma will be the same from year to year.

There were methodological issues associated with the cluster randomised trial. Although there were 12 179 children with asthma in the study, there were only 141 GP practices, which was the unit of randomisation. With 141 GP practices, there is a chance of random differences between the two intervention arms. Any random differences could be compounded by the fact that children with asthma and the common medical practice undertaken to manage asthma would tend to be more alike within practices than between practices. This may affect the clinical outcomes.

The strengths of the study were that the intervention was evaluated in a relatively large trial population of children within a primary care setting within a single year. In addition, the procedures used in the study were the same as those that would be used in clinical practice and so implementation into routine care is straightforward.

The study had a highly efficient innovative study design that used routine data for all outcomes, and the delivery of the intervention was centrally automated through the CPRD. By our own estimation, a substantial six-figure sum is saved compared with a trial where GP practices would need to be visited to collect the data. There were additional practical advantages in using routine data. For example, the planning of data collection was relatively straightforward to schedule and the collection of baseline data could be done retrospectively once practices had entered the trial.

This final strength of the trial is also a weakness. Using routine data made the assessment of unscheduled contacts within the trial difficult, especially for an intervention that increased initial medical activity through the collection of prescriptions. In this study, it would have been helpful for two additional questions to be asked to facilitate evaluation of the intervention: was the contact unscheduled? Was the contact respiratory related?

The study adds to the current literature by demonstrating that an easy to implement intervention of a simple letter from a GP to the parents of a children with asthma can assist in the self-management of the condition by increasing prescription uptake and consequently reducing medical contacts. Over 90% of medical contacts are in a primary care setting and yet there is a paucity of evaluations in this setting. This has demonstrated that using routine data collected through the CPRD within a cluster randomised trial is feasible and has shown the advantages and disadvantages of this approach.

## CONCLUSIONS

The intervention succeeded in increasing the number of children collecting a prescription in August. The intervention did not reduce unscheduled care as expected in September, which was the primary endpoint, although in the year following the intervention, it had a statistically significant, and potentially clinically important, effect on reducing total medical contacts. This is reflected in the health economic evaluation which, overall, showed that the intervention had a high probability of giving a cost saving.

With the evidence from the trial of an increase in August of both prescription collection and evidence of cost reduction practices may wish to implement the intervention, particularly practices with high rates of unscheduled medical care.

**Author affiliations**
[1]Medical Statistics Group, School of Health and Related Research (ScHARR), University of Sheffield, Sheffield, UK

[2]Health Economics and Decision Sciences, School of Health and Related Research (ScHARR), University of Sheffield, Sheffield, UK

[3]Department of General Practice, University of Cork, Cork, Ireland

[4]Department of Psychology, University of Sheffield, Sheffield, UK

[5]Respiratory Department, Sheffield Children's Hospital, Sheffield, UK

[6]Clinical Trials Research Unit, School of Health and Related Research (ScHARR), University of Sheffield, Sheffield, UK

**Acknowledgements** We gratefully acknowledge the hard work, support and advice from the following: Hilary Pinnock (Professor, University of Edinburgh) for her advice; Gerry McCann, Zaynah Gurreebun, Rachael Williams, Robin May and Jennifer Campbell, Clinical Practice Research Datalink (Clinical Practice Research Datalink, CPRD) for their contribution to site recruitment and to Tjeer Van Staa (CPRD) for his advice on CPRD; Cara Mooney and David White (University of Sheffield) for support with study set up, site recruitment and site set up and close down; Dan Beever and Helen Wakefield for administrative and clerical support; Wei Sun Kua, for her Health Economic contribution; Jonathan Boote as PPI Lead; Mike Bradburn and Neil Shephard for the statistical input and Amanda Loban for data management. We are very grateful to the children and parents who took part in the three patient and public events through all stages of the study for their advice and input. We would like to especially thank the GP adjudication panel Dr Mark Boon (Conisbrough Group Practice), Dr Karen Forshaw (Bentley Surgery), Dr Julie Hackney (The Avenue Surgery) for their advice, steer and valued contributions. We offer special thanks to the members of our oversight committee: Trial Steering Committee: Dr Steve Holmes (Independent Chair, General Practitioner), Professor Andrew Wilson (Independent Academic General Practitioner, University of Leicester), Dr Martyn Lewis (Independent Statistician, Keele University), Zaida Bibi (Independent Parent Representative) and Camilla Mills (Independent Parent Representative).

**Collaborators** Members of the PLEASANT study team: Gerry McCann, Rachael Williams, Robin May, Jennifer Campbell, Cara Mooney, David White, Dan Beever, Wei Sun Kua, Jonathan Boote, Mike Bradburn, Neil Shephard and Amanda Loban.

**Contributors** SAJ, MJH, SD and MF together produced the first draft of the report. The following conceived of or designed the work: SAJ, MJH, SD, WHS, HE, PN and CC. The following were involved in the interpretation of data for the work: SAJ, MJH, WHS and HF. The following were involved in the analysis of data: OB, SD, MF and RMS. The following drafted the paper: SAJ, MJH, SD an dMF. The following reviewed the work critically for important intellectual content. SAJ, MJH, OB, SD, MF, WHS, HE, PN and CC. All authors agree to be accountable for all aspects of the work in ensuring that questions related to the accuracy or integrity of any part of the work are appropriately investigated and resolved.

**Funding** Funding for this study was provided by the Health Technology Assessment programme of the National Institute for Health Research Programme (Grant 11/01/10). The study was sponsored by Sheffield Clinical Commissioning Group. The authors have freedom to act in the writing and submission of the paper.

**Disclaimer** The views expressed in this report are those of the authors and not necessarily those of the National Institute for Health Research Health Technology Assessment programme. Any errors are the responsibility of the authors.

**Competing interests** None declared.

**Patient consent** Not required.

**Ethics approval** Ethical approval for the study was given by South Yorkshire Research Ethics Committee on 25 October 2012 (reference number 12/YH/04).

**Provenance and peer review** Not commissioned; externally peer reviewed.

**Data sharing statement** Access to patient-level data is provided by the CPRD for health research purposes and is dependent on approval of a study protocol by the MHRA Independent Scientific Advisory Committee (ISAC). More information on ISAC and the protocol submission process can be found at: www.cprd.com/isac (date accessed 18 April 2017).

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
