## [Reviewer comments · BMJ Open]

ARTICLE DETAILS

TITLE (PROVISIONAL)	An Open-Label Cluster Randomised Controlled Trial and Economic Evaluation of a Brief Letter from a GP on Unscheduled Medical Contacts Associated with the Start of the School Year – The PLEASANT Trial
AUTHORS	Julious, Steven; Horspool, Michelle; Davis, Sarah; Franklin, Matthew; Smithson, Henry; Norman, Paul; Simpson, Rebecca; Elphick, Heather; Bortolami, Oscar; Cooper, Cindy

VERSION 1 – REVIEW

REVIEWER	Kim Lavoie University of Quebec at Montreal, Canada
REVIEW RETURNED	23-May-2017

GENERAL COMMENTS	Summary This study assessed the extent to which a public health intervention in the form of prescription reminders delivered in general practice reduced unscheduled medical contacts during a high risk period for asthma exacerbations among school aged children. Main findings were that while the intervention did increase the number of prescriptions filled by 30%, it not impact unscheduled medical contacts in September (approx. 5-6 weeks after the intervention); however it did reduce the number of medical contacts between September and December by 5%. The study also showed evidence of being cost-effective: for a cost of £1.34 it saved £36.07 per child. The authors concluded that the intervention succeeded in increasing the number of prescriptions filled and unscheduled contacts between September and December. Strengths 1. This study was very well reasoned and based on evidence showing a higher number of asthma exacerbations during the return to school period (beginning in September) due to higher exposure to viral infections, alongside a concomitant 25% drop in filling of asthma medication prescriptions. This cluster randomised trial was designed to evaluate whether a letter sent from a GP at the start of the summer vacation reminding parents of children with asthma of the necessity of taking their asthma medication before the return back to school. The study evaluated whether the letter reduced unscheduled contacts after the return back to school and increased prescriptions in August. This intervention is judged to be non-invasive, low cost, and highly feasible.2. The main study design was a cluster RCT, with a reminder letter as the active intervention and no letter as the comparison. This comparison condition, which is considered usual care, is
--

appropriate.

3. Analyses were restricted to patients with a rx of inhaled corticosteroid in the past year, which is appropriate.
4. Cluster randomization procedures were appropriate.
5. Though the study team were unblinded they were not involved in data capture and did not have access to the data until after the analysis plan was done. Adjudicators were blind to group randomization.
6. Analyses were intention to treat which is appropriate.
7. The authors appeared to have piloted their letter with a small group of parents and children which is a specific strength and follows the recommendations of the ORBIT model for behavioral intervention development (see Czajkowski et al, Health Psychology).
8. The economic analysis is judged to be a specific strength of this study.
9. The inclusion of an independent trial oversight committee is judged to be a strength.
10. The appendix including details of protocol adherence is judged to be a strength.
11. The inclusion of a CONSORT flow diagram and checklist was appropriate.

Weaknesses

1. Given that exposure to infections and their impact on asthma may take some time, it is possible that making the primary outcome period the first 4 weeks after returning to school was too soon to observe an effect of the intervention. This might explain why there was an effect when the measurement period was extended to December which may have been more appropriate to begin with.
2. Could the authors append the letter as supplemental material?
3. The authors mentioned that full details of the economic analyses will be published in a separate paper – but this is not justified. If economic analyses are included in the present paper, the methods should be described. They could then be cited in subsequent papers.
4. Could the authors confirm that the excluded practice (after providing consent) was excluded prior to randomization?
5. Page 12 seems to say that the analyses only included practices that provided data for entire study periods; does this respect the intent to treat principle?
6. Could the authors discuss some of the methodological issues with cluster RCT's, for example: how subjects within clusters tend to be more alike than those in other clusters due to shared variance within practices, and how this might affect the outcomes? This is often under-reported when discussing the limitations of cluster RCTs.
7. The authors mention adjusting for gender, but unless they measured gender using a validated measure they likely only measured biological sex (ie, whether the child was a girl or a boy) and should thus replace gender with sex throughout the manuscript.
8. On page 16, the authors concluded that “with the strong evidence from the trial of an increase in August of both prescription collection and evidence of cost reduction practices may wish to implement the intervention.” I think that the effect of this trial are generally small, and that calling them strong is overstating the results. This should be edited.
9. The appendix seemed to include the original protocol; I am not sure this is necessary.
10. The article generally reads well but there are various typos throughout that require editing, details of which go beyond the scope of this review.

REVIEWER	Dr Elora Baishnab Community Based Medical Education Division of Medical Education School of Medical Sciences Faculty of Biology Medicine and Health The University of Manchester Oxford Road Manchester M13 9PT
REVIEW RETURNED	08-Jul-2017

GENERAL COMMENTS	Excellent study with interesting results. Good patient involvement in the development of the reminder letter and timing. I was not sure whether when you referred to unscheduled medical contact you included episodes outside of GP e.g. OOH and A&E Why did you choose the QALY period as 4 months post intervention but the costs over 12 months? Did you include the costs of administrative staff time when working the latter out?
--

REVIEWER	Jacqueline Stephen Edinburgh Clinical Trials Unit, UK
REVIEW RETURNED	31-Jul-2017

GENERAL COMMENTS	Overall an interesting trial and well written paper. Some room for improvement by clarifying information required by CONSORT. Major comments 1. The abstract (p2) needs to be improved by including the information required by CONSORT. See table 2 extension of CONSORT for abstracts. Information required includes eligibility criteria for clusters, clarify whether the objective pertains to the cluster or individual level or both, how clusters were allocated to interventions, any blinding and the source of funding. In the results section, the number of participants randomised and analysed in each group need to be provided, and make clear which result is from the primary analysis. Overall, there is a need to be careful that results are not being over-interpreted and only picking the 'best' results. Only the primary outcome is required to be reported in the abstract. In particular, the conclusions state 'after September there was evidence in favour of the intervention'. The use of the word 'after' is not strictly correct due to looking at Sep, then Sep-Dec, rather than Sep and Oct-Dec. Also evidence in favour of the intervention was only found when looking at the number of total contacts Sep-Aug, no impact on unscheduled contacts for any time period or when looking at the proportion of unscheduled/total contacts. 2. Introduction (p4). Further explanation on the rationale for using a cluster design is required i.e. was there a reason why a non-cluster trial couldn't be performed? Was it used for practical reasons etc.
---

3. Research aims and objectives (p4). Clarify whether the objectives pertain to the cluster or individual participant level or both.

4. Study Design (p6). The definition of the cluster is not stated explicitly in the study design. Although it is mentioned later on in Setting (p8) it could be brought forward in the text.

The three study periods were September, September-December and September-August. It is not clear to me why the study periods were not distinct? i.e. September, October-December and January-August. This would have allowed comparisons between the distinct study periods. Since this is not the case, there is a need to ensure throughout the paper not to refer to results as 'after September' since September is included in both the additional time periods and comparisons between periods cannot be compared directly.

Appendix 2 refers to changes made to the protocol but this is not referred to in the text?

All secondary outcomes specified in the protocol need to be mentioned in the text. It can then be clarified which secondary outcomes will appear in this paper, or whether they will be looked at in another paper. Results of all secondary outcomes should be reported, and results could be given in an appendix.

It would also be beneficial to see the statistical analysis plan to see if what was pre-specified in the analyses was performed.

5. Randomisation and blinding (p8).

Details are not given on the type of randomisation, i.e. how were strata defined (what cut-point of list size of GP was chosen)? Were they then assigned to each stratum using random numbers, blocking etc?

It is not clear who assigned the cluster to intervention or control (presumably the study statistician? How was this done?). It is not clear in the methods section the mechanism by which individual participants were included in the trial? Who consented to the trial (at cluster or individual level?) and was consent obtained before or after randomisation?

6. Data management (p8). Could sensitivity analyses be performed to check the impact of the assumptions that recorded terms not coded by the adjudication panel and contacts with no terms as were defined as unscheduled contacts? Would the results be different if these were classed as scheduled contacts?

7. Analysis Populations (p9). The sample size calculation is okay but is lacking some details including ICC variability. More details are found in the protocol and should be referred to. It was not mentioned whether equal or unequal cluster sizes were assumed and given randomisation was performed stratified by cluster size suggests unequal clusters were anticipated.

8. Health economic methods (p10). I have not reviewed this section as not my area of knowledge.

9. Recruitment and participant flow (p11). Dates defining the periods of recruitment and follow-up should be provided.

10. Clinical Results (p12). A coefficient of intracluster correlation (ICC) should be given for each primary outcome. Could include in Table 2.

It was stated analyses were also performed on the per protocol population? Results should be provided (in an appendix).

Figure 2 (p33). On page 12, figure 2 is stated as the results of the incident rate ratios, however figure 2 is presenting the mean difference. I would suggest removing figure 2 as it is presenting a simple mean difference, not results from the model adjusting for covariates and clustering. It is potentially misleading.

Results should avoid being over-interpreted, in particular reference to 'there was evidence of a reduction in the mean number of medical contacts'. The initial IRR in Sep was 0.97 with 0.95 in Sep-Aug, not a huge reduction and also the added complexity that September is included in both time periods as mentioned in previous comment. This paragraph reads more as a discussion than reporting the results.

Minor Comments

1. Methods (p5)

Re-ordering of the methods section could improve the ease of reading. Perhaps begin with study design, following with participants and interventions.

2. Study Design (p6)

Make clear reference 19 is for the published trial protocol. Include 'and' after September in the following sentence: 'The effectiveness of the intervention was assessed on the basis of reduced unscheduled medical contacts after the return to school in September prescription uptake prior in August.'

3. Patient Involvement (p6).

NIHR abbreviation.

Full stop after 'There has also been a separate publication on the first two PPI consultation events'

4. Analysis populations (p9)

'The study was designed to detect a difference of 5%..' clarify this was to detect a difference of 5% in the proportion of children with unscheduled contacts.

Remove second 'and' in the following sentence 'children aged 5-16 (the primary analysis population) and children aged 5-16 and who have prescriptions for steroid preventer.'

ITT has been stated as the primary analysis, improve wording of 'The primary analyses of effectiveness were performed on both ITT with analyses also conducted on the per protocol (PP).'

Clarify what is meant by 'ITT analyses comprised all practices for whom data were obtained by study period.' It is explained better in section 4.3 (p12) where it is stated that only practices that contributed data to the entirety of that period was included.

	5. Analytical methods (p10) What is reference 23 referencing? 6. Baseline Characteristics (p11) Could you also provide descriptive statistics of the number of contacts the previous September for each trial arm since this is being used as a covariate in the analysis? 7. Clinical results (p13) It is stated 'The results are comparable for children receiving preventer medication.' Include where to find these results i.e. Table 2B. 8. Discussion (p14) Second paragraph 'they also had an increase in scheduled contacts in the same month'. Clarify this finding is for the number of scheduled contacts, not proportion. Third paragraph. 'There was an increase in the proportion of children who had an unscheduled medical contact in September'. The difference in the two proportions is small (1.5%) and not statistically significant, and therefore perhaps reading too much into this increase. Last paragraph. Re-word 'was used to definition of an unscheduled medical contact' 9. Conclusions (p16) It is stated 'the intervention succeeded in increasing the proportion of children who had scheduled contacts in the same month'. From table 3 the increase is not statistically significant (OR 1.13 (95% CI 0.84, 1.52)). 10. Table 2 (p25). Include both N (%) and Mean (standard deviation) in the table. Footnote . After 'formal statistical analysis allowing for covariates' include you have also taken into account clustering. The observed ICC should also be reported in the table. Similar comments for Table 3 (p26). 11. Figure 1 (p30) In the CONSORT diagram, clarify follow-up is for 12 months.
--	--

VERSION 1 – AUTHOR RESPONSE

[We have also uploaded the responses as supplemental information as we feel it is easier to read. Particularly the response for one question by a reviewer which was a table]

Overall Responses to the Referees and Editor

We are grateful to the reviewers and editor for their valuable comments on the paper which we feel has improved the manuscript and hopefully added greater clarity. In particular, the new Figure 2 (suggested by the second referee) has greatly enhanced the message of the paper.

The trial shows how a simple intervention can make a small (not always statistically significant) but important (and cost-effective) effect. We would hope that practices could look to implement the intervention into routine care

We also hope the trial methods used could be implemented by trialists in the primary care setting. Over 90% of medical contacts are in primary care and yet there is a paucity of trials in this setting. This point is now highlighted in the discussion along with the contribution of the trial to the literature.

Editors comments:

Q1. While the pre-specified primary outcome is reported in the paper (<http://www.isrctn.com/ISRCTN03000938mes>), there are 13 pre-specified secondary outcomes and you only report a few. Please fully report your pre-specified secondary outcomes in the manuscript.

R1. All the main endpoints have been reported in the study with respiratory contacts now added to the Appendix. The only endpoints not included are the survival analyses of the primary endpoint which were consistent with the primary endpoint.

Q2. We recommend that you remove all economic analyses and report them in a separate paper.

R2. We do feel that health economics lends to the narrative of the discussion – cheap intervention, easy to implement which is cost-saving – however, we have reduced the length of this text as the health economic analyses have already been published.

Q3. Please add a discussion of strengths and limitations in the discussion section of the paper. Similarly, please discuss what this adds to the current literature.

R3. Thank you for this suggestion. We have added text to the discussion to consider the strengths and limitations of the study.

The main strength of the study findings is that it is an easy to implement intervention that could form part of routine care planning.

The main strength of the study itself is that we used routinely collected data for the derivation of the primary endpoint. This saved a considerable sum of money. This strength was also its weakness as using routinely collected data presented data coding challenges.

Reviewer: 1

Reviewer Name
Kim Lavoie

Please leave your comments for the authors below

Summary

This study assessed the extent to which a public health intervention in the form of prescription reminders delivered in general practice reduced unscheduled medical contacts during a high risk period for asthma exacerbations among school aged children. Main findings were that while the intervention did increase the number of prescriptions filled by 30%, it not impact unscheduled medical contacts in September (approx. 5-6 weeks after the intervention); however it did reduce the number of

medical contacts between September and December by 5%. The study also showed evidence of being cost-effective: for a cost of £1.34 it saved £36.07 per child. The authors concluded that the intervention succeeded in increasing the number of prescriptions filled and unscheduled contacts between September and December.

Strengths

1. This study was very well reasoned and based on evidence showing a higher number of asthma exacerbations during the return to school period (beginning in September) due to higher exposure to viral infections, alongside a concomitant 25% drop in filling of asthma medication prescriptions. This cluster randomised trial was designed to evaluate whether a letter sent from a GP at the start of the summer vacation reminding parents of children with asthma of the necessity of taking their asthma medication before the return back to school. The study evaluated whether the letter reduced unscheduled contacts after the return back to school and increased prescriptions in August. This intervention is judged to be non-invasive, low cost, and highly feasible.
2. The main study design was a cluster RCT, with a reminder letter as the active intervention and no letter as the comparison. This comparison condition, which is considered usual care, is appropriate.
3. Analyses were restricted to patients with a rx of inhaled corticosteroid in the past year, which is appropriate.
4. Cluster randomization procedures were appropriate.
5. Though the study team were unblinded they were not involved in data capture and did not have access to the data until after the analysis plan was done. Adjudicators were blind to group randomization.
6. Analyses were intention to treat which is appropriate.
7. The authors appeared to have piloted their letter with a small group of parents and children which is a specific strength and follows the recommendations of the ORBIT model for behavioral intervention development (see Czajkowski et al, Health Psychology).
8. The economic analysis is judged to be a specific strength of this study.
9. The inclusion of an independent trial oversight committee is judged to be a strength.
10. The appendix including details of protocol adherence is judged to be a strength.
11. The inclusion of a CONSORT flow diagram and checklist was appropriate.

Weaknesses

Q1. Given that exposure to infections and their impact on asthma may take some time, it is possible that making the primary outcome period the first 4 weeks after returning to school was too soon to observe an effect of the intervention. This might explain why there was an effect when the measurement period was extended to December which may have been more appropriate to begin with.

R1. We agree with you on this point. The evidence does indicate that the peak occurs in October and November. We also made the mistake of having a dichotomised endpoint when mean contacts per child would have been better (an outcome which also ties in with the health economics). It is possible to increase the number of children who see their doctor but to then reduce the number of contacts per child.

If we were designing the trial today we would have as primary the mean number of contacts per child over the time interval September to December

To illustrate the point you raised we have included two extra figures – Figure 2a and Figure 2b - for prescription count and unscheduled medical contacts for the duration of the study. We can see here the effect of the intervention is after September with little to no effect in September for unscheduled contacts.

As a consequence of the two extra figures a new author has been added (Simpson) who created these and also conducted the additional formal statistical analyses in response to the reviews.

The results in the paper has now been rewritten to describe the effect of the intervention using these two figures in September and beyond.

For unscheduled contacts the Figure highlights the point you raised. For prescription collection there is also now seems to be an effect of increased prescriptions in September

We have also taken on board your comments more specifically in the discussion (again red is new text)

Q2. Could the authors append the letter as supplemental material?

R2. The letter is given as Appendix 1. We would be also happy for it to be attached as a separate file if the editors prefer.

Q3. The authors mentioned that full details of the economic analyses will be published in a separate paper – but this is not justified. If economic analyses are included in the present paper, the methods should be described. They could then be cited in subsequent papers.

R3. The paper has now been published and is in an open access journal. We would prefer to keep the level of detail as it is now and allow readers to refer the other article for more detail. The editor suggest that we could delete details of the economic analyses completed. However, in line with the reviewer, we believe that the analyses add to the interpretation (and main message) of the findings. Hopefully, our approach to briefly report these findings is an acceptable compromise.

Q4. Could the authors confirm that the excluded practice (after providing consent) was excluded prior to randomization?

R3. There are two forms of consent for which the trial is governed. There is the consent for the Clinical Practice Research Datalink (CPRD) to take the data from the individual GP practices (for which CPRD has an established ethics approval). There is also the consent for the GP practices to be in the PLEASANT trial. The GP practice removed their consent for CPRD to take their data.

In answer to your question therefore it was after the randomisation that the GP practice was excluded.

Q5. Page 12 seems to say that the analyses only included practices that provided data for entire study periods; does this respect the intent to treat principle?

R5. We think that it does respect the principles of ITT as the withdrawal of the practices was independent of the effect of treatment.

Practices left the study if they left the Vision IT system. The CPRD was at the time of the study tied to the GP practice computer system Vision. As we understand it the reason why the practices left Vision was due to newly formed (in England and Wales) Clinical Commissioning Groups (CCG) the GP practices joined having one system for all the practices within a CCG. If this system was not Vision it meant a practice would need to change system and leave the CPRD.

We did not foresee this happening but we believe the risk of it impacting on the results is low

Q6. Could the authors discuss some of the methodological issues with cluster RCT's, for example: how subjects within clusters tend to be more alike than those in other clusters due to shared variance within practices, and how this might affect the outcomes? This is often under-reported when discussing the limitations of cluster RCTs.

R6. We have added some text in the discussion to discuss the limitations (and strengths) of cluster RCTs. We discussed although the study had 12000 children there were only actually 141 GP practices (the unit of randomisation) and how the clinical practice within a GP practice may impact on results.

Q7. The authors mention adjusting for gender, but unless they measured gender using a validated measure they likely only measured biological sex (ie, whether the child was a girl or a boy) and should thus replace gender with sex throughout the manuscript.

R7. We have made this change throughout the paper.

Q8. On page 16, the authors concluded that "with the strong evidence from the trial of an increase in August of both prescription collection and evidence of cost reduction practices may wish to implement the intervention." I think that the effect of this trial are generally small, and that calling them strong is overstating the results. This should be edited.

R8. We have deleted the word strong.

Q9. The appendix seemed to include the original protocol; I am not sure this is necessary.

R9. A requirement for the journal is to submit the protocol with the paper. We agree it does not need to be with the paper itself but the journal can if they wish post it online with the paper

Q10. The article generally reads well but there are various typos throughout that require editing, details of which go beyond the scope of this review.

R10. We have proof read the paper again to correct these typos.

Reviewer: 2

Reviewer Name
Dr Elora Baishnab

Please leave your comments for the authors below

Q1. Excellent study with interesting results.

R1. Thank you

Q2. Good patient involvement in the development of the reminder letter and timing.

R2. Thank you

Q3. I was not sure whether when you referred to unscheduled medical contact you included episodes outside of GP e.g. OOH and A&E

R3. The unscheduled contacts does include unscheduled contacts away from the GP practice if the GP practice was informed of the contact. For children with asthma we believe that a GP practice would be informed of these contacts. The main challenge was identifying these contacts in our database – the discharge letter will more than likely be put into the GP practice systems by an administrator and then, as a result, be coded as an administrative contact.

Q4. Why did you choose the QALY period as 4 months post intervention but the costs over 12 months?

R4. The perception when designing the study was that the effect of the intervention on unscheduled contacts would be in the first 4 months. In contrast, for scheduled contacts we believed though that the effect would be to change the timing of the contacts. For example, children with asthma will have an asthma review once a year. We thus went out for 12 months to ascertain if the intervention increased scheduled care or the timing of the scheduled care i.e. the intervention moved forward scheduled medical contacts which would have happened anyway.

Q5. Did you include the costs of administrative staff time when working the latter out?

R4. We did include this time. Also we included the time taken by each practice to check the list of children with asthma identified from the CPRD as eligible for the trial.

We have added text in the method to clarify this point.

Reviewer: 3

Reviewer Name
Jacqueline Stephen

Please leave your comments for the authors below
Overall an interesting trial and well written paper. Some room for improvement by clarifying information required by CONSORT.

Major comments

Q1. The abstract (p2) needs to be improved by including the information required by CONSORT. See table 2 extension of CONSORT for abstracts.

R1. We have completed the CONSORT checklist for abstracts.

i. Information required includes eligibility criteria for clusters

R1(i). All GP practices in England and Wales were eligible to be in the trial if, at the time, they were using the Vision IT system and for whom we could access data through the CPRD. We invited practices through the CPRD to enrol in the study and this was the majority of the practices (129/142). We also invited practices through the primary care research network and these practices needed to link into CPRD to be in the trial. We have written a paper on the recruitment procedure (see below) but in the interests of space we have kept the text as is in the abstract.

Horspool MJ, Julious SA, Mooney C, May R, Sully B and Smithson WH. Preventing and lessening exacerbations of asthma in school-aged children associated with a new term (PLEASANT): recruiting primary care research sites – the PLEASANT experience. *npj Primary Care Respiratory Medicine* 2015 25, 15066; doi:10.1038/npjpcrm.2015.66

We have added some text to briefly describe recruitment (Section 3.7).

ii. Clarify whether the objective pertains to the cluster or individual level or both

R1(ii). We have added this clarification

iii. How clusters were allocated to interventions

R1(iii). We have added text on the randomisation

iv. Any blinding and the source of funding.

R1(iv). Source of funding is given below the abstract.

Within the abstract we have not stated the study is open label but we have edited the study title and the study design description in the main paper to say it is

v. In the results section, the number of participants randomised and analysed in each group need to be provided, and make clear which result is from the primary analysis.

R1(v). In the participants section we now give the number of children in the study and the number of GP practices in each arm (the unit of randomisation)

vi. Overall, there is a need to be careful that results are not being over-interpreted and only picking the 'best' results. Only the primary outcome is required to be reported in the abstract.

R1(vi). We recognise the point and we have said what the primary endpoint is throughout the abstract and the paper. See the responses to the Referee 1 there is an issue as to whether the right primary endpoint has been chosen. We are of the view that there is an effect from the intervention but we need to maintain a fine line between scientific rigour and as you state over interpretation. We feel (hopefully) that we have done this.

vii. In particular, the conclusions state 'after September there was evidence in favour of the intervention'. The use of the word 'after' is not strictly correct due to looking at Sep, then Sep-Dec, rather than Sep and Oct-Dec. Also evidence in favour of the intervention was only found when looking at the number of total contacts Sep-Aug, no impact on unscheduled contacts for any time period or when looking at the proportion of unscheduled/total contacts.

R1(vii). We will discuss time intervals later in our responses. We do feel there is an effect on unscheduled contacts. However, the effect is small. A priori we anticipated that increase prescriptions would reduce unscheduled care. We found evidence for the former but the link to medical contacts was not as strong and occurred later than we anticipated.

Q2. Introduction (p4). Further explanation on the rationale for using a cluster design is required i.e. was there a reason why a non-cluster trial couldn't be performed? Was it used for practical reasons etc.

R2. We have text to the discussion to address this point, where we discuss the strengths and limitations of the cluster design in more detail.

Q3. Research aims and objectives (p4). Clarify whether the objectives pertain to the cluster or individual participant level or both.

R3. We have added text in Section 2 to clarify that the objective pertain to contacts per child.

Q4. Study Design (p6). The definition of the cluster is not stated explicitly in the study design. Although it is mentioned later on in Setting (p8) it could be brought forward in the text.

R3. We have added text to clarify the point by stating that GP practices are randomised

Q5. The three study periods were September, September-December and September-August. It is not clear to me why the study periods were not distinct? i.e. September, October-December and January-August. This would have allowed comparisons between the distinct study periods. Since this is not the case, there is a need to ensure throughout the paper not to refer to results as 'after September' since September is included in both the additional time periods and comparisons between periods cannot be compared directly.

R5. Given the results that we have observed the time intervals as stated would have been quite useful. However, we did these intervals as we wished to see look at wider intervals but have them overlapping. This has proven useful as we feel there is evidence of increased medical activity in September – which we comment on the discussion – but this is then over weighed by the effect after then. This is important as we do not wish to have an intervention which causes more work. Including September in the assessment windows allows a determination of effect.

Given the explanations above we would prefer to keep “after” in the paper.

Q6. Appendix 2 refers to changes made to the protocol but this is not referred to in the text?

R6. We did this for good practice only. We are required to submit the ethics protocol with the paper. There can though then be important amendments which are not in the original submission. The protocol amendments for this study did not impact on the main assessments and this what the amendment highlights. We can delete if the editors see fit. Disclosure to the referees and the editors may be sufficient.

Q7. All secondary outcomes specified in the protocol need to be mentioned in the text. It can then be clarified which secondary outcomes will appear in this paper, or whether they will be looked at in another paper. Results of all secondary outcomes should be reported, and results could be given in an appendix.

R7. We have added the respiratory contacts to the Appendix. In text we have included scheduled contacts and prescriptions for the 12 months from September. We have not included the survival analysis of the primary outcome as this is consistent with the analysis reported.

Q8. It would also be beneficial to see the statistical analysis plan to see if what was pre-specified in the analyses was performed.

R8. We have included the statistical analysis plan with the revised submission.

Randomisation and blinding (p8).

Q9. Details are not given on the type of randomisation, i.e. how were strata defined (what cut-point of list size of GP was chosen)? Were they then assigned to each stratum using random numbers, blocking etc?

It is not clear who assigned the cluster to intervention or control (presumably the study statistician? How was this done?). It is not clear in the methods section the mechanism by which individual participants were included in the trial? Who consented to the trial (at cluster or individual level?) and was consent obtained before or after randomisation?

We have said now it is block randomisation and have added text to the method to clarify points about the process for the randomisation and who consented.

We have also added reference to the paper

Horspool, MJ, Julious, SA, Mooney, C, May, R, Sully, B, Smithson, W. Preventing and Lessening Exacerbations of Asthma in School-aged children Associated with a New Term (PLEASANT): Recruiting Primary Care Research Sites-the PLEASANT experience. NPJ Prim. care Respir. Med. 25, 15066 (2015)

In this section which describes the recruitment process for the trial

The strata sizes for the practices used in the randomisation were

1. <3072
2. 3072-4561
3. 4562-5564
4. 5565-6645
5. 6646-7933
6. 7934-9242
7. 9243-10264
8. 10265-11661
9. 11662-14211
10. >14211

Q10. Data management (p8). Could sensitivity analyses be performed to check the impact of the assumptions that recorded terms not coded by the adjudication panel and contacts with no terms as were defined as unscheduled contacts? Would the results be different if these were classed as scheduled contacts?

R10. This is an interesting point. These account for around 15% of contacts in total. With respect to the free text, the advice from the adjudication panel was that if a contact was recorded as free text it would be because the GP would be too busy to code the contact and would making quick notes but the meeting would be unscheduled. If a prescription was associated with the contact, we could also assign this to respiratory related.

In addition the above scheduled contacts have particularly codes which are associated with them and annual asthma reviews are part of the care plan for children with asthma.

It obviously is less clear for the contacts not adjudicated on

If it is OK we would like to look at this outside of the paper.

If coded or not these contacts are associated with a medical contact. Our own point of view for a sensitivity analysis would be to err to omitting them but the point made about scheduled is a good point.

To a limited degree the endpoint total contacts picks up a little on these data.

Q11. Analysis Populations (p9). The sample size calculation is okay but is lacking some details including ICC variability. More details are found in the protocol and should be referred to. It was not mentioned whether equal or unequal cluster sizes were assumed and given randomisation was performed stratified by cluster size suggests unequal clusters were anticipated.

R11. We have added a reference to the protocol and mentioned about the sensitivity analysis. We did assume equal cluster size in the study design and we state this now.

We did do sensitivity analyses for the practice size but the impact on the power was small. At the start of the study the main concerns were the assumptions we made for the ICC and how many practices would not send the intervention.

Q12. Health economic methods (p10). I have not reviewed this section as not my area of knowledge.

R12. This has now successfully gone through peer review and has been published in a specialist open access journal

Q13. Recruitment and participant flow (p11). Dates defining the periods of recruitment and follow-up should be provided.

R13. Recruitment of GP practices was undertaken over a 7-month period. Details of which have been published 21. We have stated what the recruitment window now is in the paper

Q14. Clinical Results (p12). A coefficient of intracluster correlation (ICC) should be given for each primary outcome. Could include in Table 2.

R14. We have added a sentence for the ICC for the primary outcome.

Q15. It was stated analyses were also performed on the per protocol population? Results should be provided (in an appendix).

R15. The paper is already quite long and we would prefer not to include the per protocol results. We have included other endpoints in the appendix not previously included in the appendix. The full trial results are in the HTA report for the study

Julious, S.A., Horspool, M.J., Davis, S. et al. (2016) PLEASANT: Preventing and Lessening Exacerbations of Asthma in School-age children Associated with a New Term - a cluster randomised controlled trial and economic evaluation. Health Technology Assessment, 20 (93). pp. 1-154. ISSN 1366-5278

Q16. Figure 2 (p33). On page 12, figure 2 is stated as the results of the incident rate ratios, however figure 2 is presenting the mean difference. I would suggest removing figure 2 as it is presenting a simple mean difference, not results from the model adjusting for covariates and clustering. It is potentially misleading.

R16. This figure has now been deleted.

Q17. Results should avoid being over-interpreted, in particular reference to 'there was evidence of a reduction in the mean number of medical contacts'. The initial IRR in Sep was 0.97 with 0.95 in Sep-Aug, not a huge reduction and also the added complexity that September is included in both time periods as mentioned in previous comment. This paragraph reads more as a discussion than reporting the results.

R17. The results section has been rewritten and a new Figure 2 has been included to show the figure giving the time course. The text now starts with summarising the unscheduled contacts (which is consistent with Figure 2). Information on the number of contacts per child has been deleted.

We think the effects, though small, are real and important As this is a low cost intervention and, as the result of the trial indicate, even a small effect are likely to be cost effective

Minor Comments

Q1. Methods (p5)

Re-ordering of the methods section could improve the ease of reading. Perhaps begin with study design, following with participants and interventions.

R1. Done.

Q2. Study Design (p6)

Make clear reference 19 is for the published trial protocol.

Include 'and' after September in the following sentence: 'The effectiveness of the intervention was assessed on the basis of reduced unscheduled medical contacts after the return to school in September prescription uptake prior in August.'

R2. Done.

Q3. Patient Involvement (p6).

NIHR abbreviation.

Full stop after 'There has also been a separate publication on the first two PPI consultation events'

R3. Done.

Q4. Analysis populations (p9)

'The study was designed to detect a difference of 5%..' clarify this was to detect a difference of 5% in the proportion of children with unscheduled contacts.

Remove second 'and' in the following sentence 'children aged 5-16 (the primary analysis population) and children aged 5-16 and who have prescriptions for steroid preventer.'

ITT has been stated as the primary analysis, improve wording of 'The primary analyses of effectiveness were performed on both ITT with analyses also conducted on the per protocol (PP).'

Clarify what is meant by 'ITT analyses comprised all practices for whom data were obtained by study period.' It is explained better in section 4.3 (p12) where it is stated that only practices that contributed data to the entirety of that period was included.

R4. All done.

5. Analytical methods (p10)

What is reference 23 referencing?

R5. It was supposed to be the protocol which has now been corrected.

6. Baseline Characteristics (p11)

Could you also provide descriptive statistics of the number of contacts the previous September for each trial arm since this is being used as a covariate in the analysis?

Variable
September

Letter (N=5917)

Mean (SD)
0.84 (1.20)
Median (IQR)
0 (0 - 1)
Range
0 - 10

No Letter (N=6262)

Mean (SD)
0.88 (1.26)
Median (IQR)
0 (0 - 1)
Range
0 - 12

Total (N=12179)

Mean (SD)
0.86 (1.23)
Median (IQR)
0 (0 - 1)
Range
0 - 12

September to December

Letter (N=5917)

Mean (SD)

3.65 (3.41)

Median (IQR)

3 (1 - 5)

Range

0 - 31

No Letter (N=6262)

Mean (SD)

3.78 (3.66)

Median (IQR)

3 (1 - 5)

Range

0 - 51

Total (N=12179)

Mean (SD)

0.86 (1.23)

Median (IQR)

3 (1 - 5)

Range

0 - 51

September to August

Letter (N=5917)

Mean (SD)

10.16 (8.14)

Median (IQR)

8 (5 - 13)

Range

0 - 74

No Letter (N=6262)

Mean (SD)

10.37 (8.72)

Median (IQR)

8 (5 - 14)

Range

0 - 126

Total (N=12179)

Mean (SD)

10.27 (8.45)
Median (IQR)
8 (5 - 14)
Range
0 - 126

Q7. Clinical results (p13)

It is stated 'The results are comparable for children receiving preventer medication.' Include where to find these results i.e. Table 2B.

R7. Done.

Q8. Discussion (p14)

Second paragraph 'they also had an increase in scheduled contacts in the same month'. Clarify this finding is for the number of scheduled contacts, not proportion.

R8. Done.

Q9. Third paragraph. 'There was an increase in the proportion of children who had an unscheduled medical contact in September'. The difference in the two proportions is small (1.5%) and not statistically significant, and therefore perhaps reading too much into this increase.

R9. It is small but we feel to be plausible given what we have discussed above. We feel there has been an initial increase in medical activity which is then offset by subsequent falls as children have had prescriptions and maybe medicines reviews if they have not recently collected preventers.

Q10. Last paragraph. Re-word 'was used to definition of an unscheduled medical contact'

R10. Done.

Q11. Conclusions (p16)

It is stated 'the intervention succeeded in increasing the proportion of children who had scheduled contacts in the same month'. From table 3 the increase is not statistically significant (OR 1.13 (95% CI 0.84, 1.52)).

R11. We feel the effect is real, even if small.

Q12. Table 2 (p25).

Include both N (%) and Mean (standard deviation) in the table.

Footnote . After 'formal statistical analysis allowing for covariates' include you have also taken into account clustering.

The observed ICC should also be reported in the table.

Similar comments for Table 3 (p26).

R12. We have given the ICC for the primary endpoint in the text.

We have added that the effect of clustering is accounted for.

We would prefer not to include the SDs and Ns to stop the table being too cluttered. We have, however, included the summary numbers to aid in the interpretation of the ORs and IRRs.

Q13. Figure 1 (p30)

In the CONSORT diagram, clarify follow-up is for 12 months.

R13. Done.

VERSION 2 – REVIEW

REVIEWER	Kim Lavoie UQAM, Canada
REVIEW RETURNED	13-Oct-2017

GENERAL COMMENTS	Overall, the authors have been very responsive to reviewers' comments and the manuscript has been much improved. I only have a few minor outstanding issues:  1. Given that some practices were excluded after randomization, they should technically be included in ITT analyses. However, the authors state that practices were excluded (all in the intervention arm) when GPs withdrew from their practices from the CPRD system (Vision IT system), which was necessary for data collection in the Pleasant Trial. This seems like an inclusion criteria, so if GPs were not connected to the CPRD/Vision IT system they could not participate. But given that these practices were randomized (and that only the intervention arm ended up with practices being excluded) then I suggest running sensitivity analyses to ensure the excluded practices were not different from those included in any systematic way. 2. In the marked up copy, the edited in portions were almost devoid of commas, making the language very difficult to read and follow. Once again, the language quality of the manuscript needs revising.
--

REVIEWER	Jacqueline Stephen Edinburgh Clinical Trials Unit (ECTU) The Usher Institute of Population Health Sciences and Informatics, University of Edinburgh UK.
REVIEW RETURNED	30-Oct-2017

GENERAL COMMENTS	Thanks for your thorough response to reviewer comments. Major comments  1. I still feel there is a need to clarify when reporting 'there was evidence in favour of the intervention' that this is a potential small clinically important effect and not a statistically significant one. It is possible that the real effect is smaller (or larger) than the observed point estimate. Leave to the editor's decision.
--

2. I still feel strongly that the results should not be reported as 'after September' since September is included in both the additional time periods and comparisons between periods cannot be compared directly.

Your response was 'This has proven useful as we feel there is evidence of increased medical activity in September – which we comment on the discussion – but this is then over weighed by the effect after then. This is important as we do not wish to have an intervention which causes more work. Including September in the assessment windows allows a determination of effect.' There is no formal analyses on whether the effect was 'over weighed by the effect' after September, the results look at September and also September to December.

The final decision should lie with the editors.

3. I think the per protocol results for the primary analysis should be included in the paper, at least a statement whether the results did or did not reach the same conclusions.

4. Section 4.4, Clinical Results.

The results section, now re-written, focusses too much on the raw data in figure 2.

There is no indication of variation on the graph, and stating 'but now after the intervention has been sent there is seems to be no immediate effect of the intervention in September and the peak in October/November is less pronounced than compared to the no letter arm.' Seems to be over-interpreting somewhat. If you had error limits around your point estimates, the difference might not be so obvious. Secondly these comparisons are different to the study periods you have specified, i.e. looking at October and November distinctly rather than the Sep-Dec combined.

The results should be as laid out in the SAP, and any post-hoc analyses clearly labelled as such.

Minor Comments

1. Baseline Characteristics

Could the descriptive statistics of the number of contacts the previous September for each trial arm be included in the actual Table 1 since this is being used as a covariate in the analysis?

2. The analysis of respiratory relation contacts

It would be worth reporting the increase in unscheduled respiratory related medical contacts and commenting on in the discussion.

3. Could you include statement in the paper to say survival analyses were consistent with the primary endpoint?

From the HTA report, I can see a statistically significant effect for the survival analysis outcome, time to first respiratory-related unscheduled contact (HR 1.30 (1.03 to 1.64)). Is this consistent with the primary outcome?

4. Discussion

	Third paragraph. 'There was an increase in the proportion of children who had an unscheduled medical contact in September'. The difference in the two proportions is small (1.5%) and not statistically significant. Your reply stated 'It is small but we feel to be plausible given what we have discussed above'. I think it should be made clear that the increase was not statistically significant and perhaps refer to it as a small but clinically important increase? The confidence interval does not exclude 0, and therefore this is also a plausible effect. Leave to the editor's decision. 5. Conclusions It is stated 'the intervention succeeded in increasing the proportion of children who had scheduled contacts in the same month'. From table 3 the increase is not statistically significant (OR 1.13 (95% CI 0.84, 1.52)). Your reply 'We feel the effect is real, even if small.' Again, I think this should be emphasised as a clinically important effect. Leave to the editor's decision.
--	---

VERSION 2 – AUTHOR RESPONSE

We are grateful for the comments of the reviewers which we have taken on board in the revised submission which we think is improved for the revisions.

We have detailed the responses to the reviewer comments below

Steven

Edit made but not asked for

We have changed the title so it refers to unscheduled medical contacts given that reducing these contacts (rather than asthma episodes) is the main objective of the trial

Editorial Comments:

Q1. Regarding reviewer 2's comments: we are happy with your response to point two below but please can you revise your manuscript to address the reviewer's first point ("I still feel there is a need to clarify when reporting 'there was evidence in favour of the intervention' that this is a potential small clinically important effect and not a statistically significant one.")

Likewise, we would like you to revise the paper as reviewer 2 suggests below:

Q2 Discussion: "I think it should be made clear that the increase was not statistically significant and perhaps refer to it as a small but clinically important increase? The confidence interval does not exclude 0, and therefore this is also a plausible effect."

Conclusions: "Again, I think this should be emphasised as a clinically important effect."

R1 and R2. We have amended the text throughout the manuscript to make the direction and significance of the effects clearer. We also discuss the clinical importance of some of the effects.

Reviewer Name: Kim Lavoie

Q1. Given that some practices were excluded after randomization, they should technically be included in ITT analyses. However, the authors state that practices were excluded (all in the intervention arm) when GPs withdrew from their practices from the CPRD system (Vision IT system), which was necessary for data collection in the Pleasant Trial. This seems like an inclusion criteria, so if GPs were not connected to the CPRD/Vision IT system they could not participate. But given that these practices were randomized (and that only the intervention arm ended up with practices being excluded) then I suggest running sensitivity analyses to ensure the excluded practices were not different from those included in any systematic way.

R1. We are able to clarify this issue with respect to the respect to GP practices withdrawing as follows:

- a. At the time of the study, if GP practices left the Vision system then they also left CPRD. However, we would have data up to the point that they left. Thus, if a practice left on April 2nd 2014 we would have data for that practice until 1st April 2014.
- b. One practice in the control arm left the CPRD and also withdrew their consent for the data from the practice to be used until the time-point they left. As a result, we could not include them in the analysis.
- c. Practices in the intervention arm sent out the letter instructing parents of children with asthma about the importance of taking preventer medication. The CPRD identified the patients and sent the list to the GP practices for them to send out the intervention on their letter headed paper. The GP practices reviewed the list prior to sending out the letter and removed certain patients from the list. The CPRD identified patients were used for the ITT analysis.

We hope the above has clarified the reviewer's query.

Q2. In the marked up copy, the edited in portions were almost devoid of commas, making the language very difficult to read and follow. Once again, the language quality of the manuscript needs revising.

R2. We apologise for this. We have gone through the manuscript and have sought to improve the quality of the language (and punctuation).

Reviewer Name: Jacqueline Stephen

Major comments

Q1. I still feel there is a need to clarify when reporting 'there was evidence in favour of the intervention' that this is a potential small clinically important effect and not a statistically significant one.

It is possible that the real effect is smaller (or larger) than the observed point estimate.

R1. As outlined in our response to the editor, we have amended the text throughout the manuscript to make the direction and statistical significance of the effects clearer. We also discuss the clinical importance of some of the effects..

Q2. I still feel strongly that the results should not be reported as 'after September' since September is included in both the additional time periods and comparisons between periods cannot be compared directly.

Your response was 'This has proven useful as we feel there is evidence of increased medical activity in September – which we comment on the discussion – but this is then over weighed by the effect after then. This is important as we do not wish to have an intervention which causes more work. Including September in the assessment windows allows a determination of effect.' There is no formal analyses on whether the effect was 'over weighed by the effect' after September, the results look at September and also September to December.

R2. The analysis below is the analysis requested which has the September and September-December results as well as the estimates for October-December

	September	October-December	September-December
Unscheduled contacts	1.01 (0.92 to 1.10)	0.97 (0.90 to 1.05)	0.97 (0.92 to 1.03)
Total Contacts	0.96 (0.86 to 1.07)	0.95 (0.88 to 1.03)	0.95 (0.89 to 1.01)

We have sought to address the reviewer's original point by removing all references to 'after September' in the paper and replacing the text with a clearer indication of the appropriate time period – e.g. "September-December" or "the year following the intervention".

R3. I think the per protocol results for the primary analysis should be included in the paper, at least a statement whether the results did or did not reach the same conclusions.

Q3 We have added a sentence to comment on the per protocol results which were in line with the ITT analysis.

Q4. Section 4.4, Clinical Results.

The results section, now re-written, focusses too much on the raw data in figure 2.

There is no indication of variation on the graph, and stating 'but now after the intervention has been sent there is seems to be no immediate effect of the intervention in September and the peak in October/November is less pronounced than compared to the no letter arm.' Seems to be over-interpreting somewhat. If you had error limits around your point estimates, the difference might not be so obvious. Secondly these comparisons are different to the study periods you have specified, i.e. looking at October and November distinctly rather than the Sep-Dec combined.

The results should be as laid out in the SAP, and any post-hoc analyses clearly labelled as such.

R4. We appreciate the reviewer's comment of the presentation of the results. We have therefore changed the order of the results section so that we now report the statistics first before referring to the figure to illuminate the findings. Narratively, we feel the figures are important for two reasons. First, they help to describe the seasonal patterns within the data for both prescriptions and medical contact. Second, they help to explain the lack of an effect in September and then the effect in the wider time intervals which include September.

Minor Comments

Q6. Baseline Characteristics

Could the descriptive statistics of the number of contacts the previous September for each trial arm be included in the actual Table 1 since this is being used as a covariate in the analysis?

R6. As requested, we have added these data to Table 1.

Q7. The analysis of respiratory relation contacts

It would be worth reporting the increase in unscheduled respiratory related medical contacts and commenting on in the discussion.

R7. We do not feel the intervention has increased respiratory contacts. There are not many respiratory contacts (relatively) and this type of contact is sensitive to whether or not a prescription was collected in the medical contact. If there was nothing else to code the contact but a prescription was collected – the request of which may have been prompted by the intervention – the contact would be coded as unscheduled.

In terms of total number of respiratory contacts the groups are similar but then a greater proportion of these contacts are recorded as unscheduled in the intervention arm.

Q8. Could you include statement in the paper to say survival analyses were consistent with the primary endpoint?

From the HTA report, I can see a statistically significant effect for the survival analysis outcome, time to first respiratory-related unscheduled contact (HR 1.30 (1.03 to 1.64)). Is this consistent with the primary outcome?

R8. Please see our comment for respiratory contacts above. Our preference would be not to comment on the HR results as they are consistent with the results we have presented in the paper.

4. Discussion

Q9. Third paragraph. 'There was an increase in the proportion of children who had an unscheduled medical contact in September'. The difference in the two proportions is small (1.5%) and not statistically significant.

Your reply stated 'It is small but we feel to be plausible given what we have discussed above'.

I think it should be made clear that the increase was not statistically significant and perhaps refer to it as a small but clinically important increase?

The confidence interval does not exclude 0, and therefore this is also a plausible effect.

R9. As outlined in our response to the editor and the other reviewer, we have amended the text throughout the manuscript to make the direction and significance of the effects clearer. We also discuss the clinical importance of some of the effects..

Q10. Conclusions

It is stated 'the intervention succeeded in increasing the proportion of children who had scheduled contacts in the same month'. From table 3 the increase is not statistically significant (OR 1.13 (95% CI 0.84, 1.52)).

Your reply 'We feel the effect is real, even if small.'

Again, I think this should be emphasised as a clinically important effect.

R10. Please see our response to Q9.

VERSION 3 – REVIEW

REVIEWER	Jacqueline Stephen Edinburgh Clinical Trials Unit (ECTU) The Usher Institute of Population Health Sciences and Informatics, University of Edinburgh UK.
REVIEW RETURNED	18-Dec-2017
GENERAL COMMENTS	Thanks for taking on board my reviewer comments, and the additional improvements in language allows for much easier reading. One final comment, all secondary outcomes need to be reported in the paper as required by CONSORT and also highlighted by the editor after the first submission. The complete list of secondary outcomes should be given in the methods section, and the results of the survival analyses (and any other outcomes not yet reported) should be included in the appendix as you have done for respiratory relation contacts. This will complete the reporting of the trial.
REVIEWER	Kim Lavoie UQAM, Canada
REVIEW RETURNED	08-Jan-2018
GENERAL COMMENTS	I am satisfied with the authors' final revision and believe the ms is now suitable for publication in BMJ Open.